# A Social Cognitive Assessment of Workplace Sedentary Behavior among a Sample of University Employees

**DOI:** 10.3390/ijerph20156476

**Published:** 2023-07-31

**Authors:** Amanda H. Wilkerson, Nuha Abutalib, Ny’Nika T. McFadden, Shristi Bhochhibhoya, Adriana Dragicevic, Bushra R. Salous, Vinayak K. Nahar

**Affiliations:** 1Department of Health Science, The University of Alabama, Tuscaloosa, AL 35401, USA; nabutaleb@crimson.ua.edu (N.A.); ntmcfadden@crimson.ua.edu (N.T.M.); 2Center for Research on Interpersonal Violence, Georgia State University, Atlanta, GA 30302, USA; shristib12@gmail.com; 3Department of Health and Exercise Science, The University of Oklahoma, Norman, OK 73019, USA; dragicevicadriana3@gmail.com (A.D.); bushra.r.salous@ou.edu (B.R.S.); 4Department of Dermatology, School of Medicine, University of Mississippi Medical Center, Jackson, MS 39216, USA; vnahar@umc.edu; 5Department of Preventive Medicine, John D. Bower School of Population Health, University of Mississippi Medical Center, Jackson, MS 39216, USA

**Keywords:** adults, worksite, sedentary behavior, social cognitive theory, workplace

## Abstract

Sedentary behavior (SB) is an important public health concern. Adults working in desk-based occupations spend a considerable proportion of the workday sitting. More information is needed regarding the factors that contribute to occupational SB. The aim of this study was to assess the utility of social cognitive theory (SCT) to explain work-related SB using a quantitative, cross-sectional design by administering an online questionnaire. Participants included 381 full-time employees at a large, public university in the south-central United States. Hierarchical multiple linear regression was used to determine the relationship between SCT constructs and SB. Mean work-related SB was 5.95 (SD = 1.30) h/8 h workday. In model 1, 9.6% of the total variance in SB was accounted for by standing desk ownership and physical activity level (*p* = 0.001; R2 = 0.096). In model 2, SCT constructs led to a statistically significant R^2^ increase of 4.9% (*p* < 0.001), where standing desk ownership, physical activity, and self-efficacy explained 13.3% of the variance in work-related SB. Findings from this study suggest that self-efficacy may be an important factor in explaining variation in occupational SB. Public health researchers and practitioners should consider strategies to address self-efficacy when developing workplace interventions to target occupational SB.

## 1. Introduction

Sedentary behavior is a significant public health concern [1,2]. Sedentary behavior refers to engagement in activities that result in low energy expenditure (i.e., 1.5 metabolic equivalents [METs]) for an extended period, such as TV viewing and computer work [3]. Consequently, increased time spent in sedentary behavior is associated with risk for chronic health conditions such as diabetes, hypertension, and cardiovascular disease [3,4,5]. Recent epidemiological evidence suggests a dose–response relationship between sedentary behavior and all-cause, as well as cardiovascular disease, mortality, where individuals who engage in high levels of sedentary behavior and low levels of physical activity are at the highest mortality risk [5,6]. The recent 2018 update of the Physical Activity Guidelines for Americans emphasizes the importance of not only engaging in physical activity but also encourages adults aged 18–64 to “move more and sit less” throughout the day [7]. The World Health Organization (WHO) recommends that adults aged 18–64 should limit time spent being sedentary and replace sedentary time with any intensity of physical activity [8]. The WHO also emphasizes that adults should aim to exceed the recommended amounts of physical activity to reduce the negative health impacts that result from high levels of sedentary behavior [8].

One of the primary domains to target sedentary behavior among adults is in the workplace [2]. Research estimates that adults working in desk-based occupations spend anywhere from 66% to 82% of their work time sedentary [9]. To date, workplace interventions have demonstrated efficacy in reducing sedentary behavior among working adults [10,11]. Most workplace interventions that have successfully reduced sedentary behavior have targeted environmental determinants of behavior through modifications of the workspace, primarily using standing or height-adjustable desks that encourage standing behavior and reduce extended bouts of sedentary behavior [10]. Evidence from workplace interventions supports the success of environmental strategies to reduce sedentary behavior, with reductions in daily sedentary behavior ranging from 15 to 89 min post-intervention [10]. In addition to environmental strategies, a recent review suggests that multi-component interventions incorporating environmental change strategies (i.e., height-adjustable or standing desks) with educational and behavioral strategies are most effective [10,11]. More information regarding appropriate educational and behavioral targets is needed to help inform the development of effective multi-component workplace interventions.

Currently, there is minimal information available regarding the theoretical antecedents of health behavior change to reduce sedentary behavior during the workday. Previous cross-sectional, theory-based studies have explored sedentary behavior among working adults primarily using the theory of planned behavior (TPB) [12,13,14,15] and social ecological model (SEM) [16,17]. Studies using the TPB have found attitude and subjective norms to be indirect determinants of sedentary behavior, where intention and perceived behavioral control have been shown to be direct determinants [12,13,14,15]. Recent studies exploring social ecological factors associated with workplace sedentary behavior using the SEM have aimed to identify additional, multi-level factors associated with workplace sedentary behavior among employees in desk-based occupations. Research in this area has shown that workplace built environmental factors, such as workplace connectedness and office type (i.e., private versus shared), may be associated with workplace sedentary behavior in addition to psychosocial factors identified in previous research [16,17]. Although previous research has applied theoretical frameworks to identify factors associated with workplace sedentary behavior, the two frameworks may not identify all factors associated with workplace sedentary behavior, such as social and built environmental factors in addition to psychosocial and cognitive factors.

Theory-based interventions that address sedentary behavior in workplace settings are limited [18]. Recent multi-component workplace interventions have included theoretical frameworks in intervention design and evaluation and have demonstrated success in reducing sedentary behavior post-intervention [19]. Further exploration of theoretical constructs associated with sedentary behavior change may provide additional intervention targets to supplement and enhance current multi-component interventions and develop effective educational and behavioral intervention components [20,21]. Among the theory-based, multi-component interventions implemented to date, most have been informed by social cognitive theory (SCT) [18]. SCT posits that engagement in a health behavior is a result of the reciprocal relationship among cognitive perceptions, social/physical environment, and health behavior, reminiscent of the intervention targets in multi-component interventions [22]. However, the role of theoretical factors in reducing sedentary behavior is unclear and requires further exploration [18]. Therefore, the purpose of this study was to conduct an exploratory analysis to determine the utility of SCT constructs to explain occupational sedentary behavior among a sample of university employees. Determining the association between SCT constructs and workplace sedentary behavior may help researchers and practitioners determine how intervention techniques can specifically address behavioral change to increase future intervention effectiveness.

### Theoretical Framework

The present study was informed by SCT [22]. SCT posits that the regulation of human behavior occurs through a comprehensive structure where an individual’s cognitive perceptions, environment, and health behaviors influence one another in a reciprocal nature, illustrated by reciprocal determinism. In SCT, self-efficacy perceptions work together with goals and objectives, expectations of results (e.g., outcome expectations), and perceived environment in influencing health behavior [22]. In this regard, the knowledge of health benefits associated with reducing sedentary behavior and self-efficacy perceptions that people can take control over their health behaviors are motivating factors that work together and encourage people to decide to reduce their sedentary behavior [22]. SCT also suggests that the comparison between the expected health outcomes of reducing sitting time versus continuing to engage in sustained sedentary behavior encourages people to change their behavior despite the cost associated with the endeavor. Understanding health risks and perceptions are the precursors to changing health behavior [22]. SCT and its associated constructs can be applied to encourage people to avoid sedentary behavior in the workplace and inform the development of multi-component workplace interventions.

Several SCT variables were operationalized in the present study. Self-efficacy was operationalized as the employee’s confidence in their ability to reduce sedentary behavior during the workday. Situational perception, or the social environment, in the present study was defined as employees’ perception of the workplace social environment (e.g., social norms) to decrease occupational sedentary behavior. The physical environment was operationalized as the employees’ perception of their physical office environment related to active living in the workplace (i.e., using stairs, ambulating around the workplace). Outcome expectations was defined as employees’ anticipation of the physical, social, and self-evaluative outcomes of reducing sedentary behavior during the workday. Lastly, outcome expectancies were operationalized as the value a person feels about the consequences (e.g., outcome expectations) of reducing sedentary behavior during the workday [22].

## 2. Materials and Methods

### 2.1. Sample and Recruitment

This study used a cross-sectional, quantitative research approach using a 57-item online survey administered with Qualtrics^TM^ online survey software. All data were collected from March to April 2019. Participants included non-instructional employees (i.e., hourly and salaried staff) at a large, public university in the south-central United States. Participants were included in the study if they were: (1) full-time, non-instructional employees; (2) working in a desk-based job (i.e., job requiring ≥ 4 h of sitting per day on average); and (3) at least 18 years old at the time of study participation. Screening questions were included at the beginning of the online survey. Participants who did not meet screening criteria were not permitted to complete the survey. A total of 422 participants completed any part of the online survey. Prior to analysis, participants with large amounts of missing data for most study variables were removed from the analytical sample. Subsequent missing data were subject to listwise deletion. After reviewing for missing data, 41 participants were excluded from the analysis. Thus, the final analytical sample size included 381 participants.

Recruitment of participants occurred through the university’s employee wellness program. The director of the wellness program assisted the research team with recruitment by sending a recruitment email to all full-time employees at the university as well as posting the recruitment message on the online wellness portal. Employees were offered 300 “wellness points” in the university’s wellness incentive program in exchange for participation in the study. The recruitment message contained information about the study, contact information for the PI, and a link to the online survey. The first page of the survey contained all informed consent information. Participants were instructed that clicking “Yes, I agree to participate” to start the survey and move past the first page of the survey indicated informed consent for participation. All recruitment and study procedures were approved by the university’s Institutional Review Board (IRB; Protocol 9220) prior to recruitment and data collection.

### 2.2. Measures

The 57-item online survey included demographic items, items to assess social cognitive theory constructs, and self-reported physical activity and occupational sedentary behavior. Nine demographic items were included in the survey. The first three demographic items were used to assess study eligibility and used forced completion. Screening items included age, employment status (i.e., full-time or 1.0 FTE), and employment in a desk-based occupation (i.e., ≥4 h of sitting per day). Participants who failed to meet the inclusion criteria were not permitted to continue the survey. The remaining six demographic items assessed participants’ self-reported race, ethnicity, gender identity, marital status, education level, and job category using the university’s job category descriptors (i.e., staff paid monthly, staff paid hourly). The remaining items were used to assess theoretical constructs, work-related factors, and self-reported behavior.

#### 2.2.1. Self-Efficacy

In the present study, self-efficacy was operationalized as the participants’ confidence in their ability to reduce their sedentary behavior during the workday. Self-efficacy to reduce sedentary behavior during the workday was assessed using an adapted version of the 8-item Self-Efficacy for Exercise Scale (SEE). The original measure has well-established internal consistency reliability (Cronbach’s α = 0.92) and predictive validity for exercise behavior in older adult populations in the United States [23]. The SEE has been modified and used in previous occupational sedentary behavior research and has demonstrated sufficient internal consistency reliability among a Flemish sample of working adults (Cronbach’s α = 0.81) [24]. The modified version of the scale adapted for use in sedentary behavior research was used in the present study. Responses in the self-efficacy scale were rated on a 5-point Likert scale (“not at all confident” = 1; “very confident” = 5). Responses to each item were summed to generate a total scale score. Total scores ranged from 5 to 40, with higher scores indicating higher self-efficacy to reduce sedentary behavior during the workday. An example item from the scale states, “How confident are you that you could reduce your sitting time each workday if you felt tired?” In the present study, the scale demonstrated sufficient internal consistency reliability (Cronbach’s α = 0.90).

#### 2.2.2. Situational Perception

Situational perception was operationalized in the present study as the employees’ perception of the workplace social environment (i.e., workplace social norms) as it relates to reducing occupational sedentary behavior. An 8-item scale that was previously validated in a sample of working adults in Australia (Cronbach’s α = 0.81) was used to assess perceived organizational social norms [25]. Items were rated on a 5-point Likert scale (“strongly disagree” = 1; “strongly agree” = 5). Items were summed to generate a total scale score, with total scores ranging from 5 to 40 and higher scores indicating higher perceived organizational social norms related to reducing sedentary behavior. A sample item from the scale states, “My workplace is committed to supporting staff choices to stand or move more at work.” The scale demonstrated sufficient internal consistency reliability in the present study (Cronbach’s α = 0.80).

#### 2.2.3. Environment

Environment in the present study was operationalized as the employees’ perception of their physical office environment as it relates to active living in the workplace. The 3 items in the perceived indoor environment sub-scale from the Supportive Environments for Active Living Survey (SEALS) were used to measure the work environment [26]. The perceived indoor environment sub-scale in the SEALS has demonstrated acceptable internal consistency reliability in previous research among a sample of working adults in the United States (Cronbach’s α = 0.79) [26]. Items are measured on a 4-point Likert scale (“strongly disagree” = 1; “strongly agree” = 4). A total scale score was calculated by summing the responses to each item with possible scores ranging from 3 to 12. A sample item states, “In general, the stairs at my worksite are accessible (i.e., no doors or doors remain unlocked)”. The perceived indoor environment sub-scale from the SEALS was found to have sufficient internal consistency reliability in the present study (Cronbach’s α = 0.85).

#### 2.2.4. Outcome Expectations and Outcome Expectancies

Outcome expectations and outcome expectancies were each measured with 9 items. Each outcome expectation item assessed participants’ agreement using a 5-point Likert scale (“strongly disagree” = 1; “strongly agree” = 5) that certain physical, self-evaluative, and social outcomes would result from reducing sedentary behavior. Outcome expectation items were modified from the Multidimensional Outcomes Expectations for Exercise Scale (MOEES), which has previously demonstrated appropriate reliability in a sample of older adults in the United States (Cronbach’s α = 0.81 to 0.84) [27]. Items in the MOESS were modified to reflect outcomes associated with sedentary behavior in the literature [28]. A paired outcome expectancy item was included for each outcome expectation item to assess participants’ valuation of that outcome expectation and was measured on a 5-point Likert scale (“not important” = 1; “very important” = 5). An example outcome expectation item states, “Sitting less each workday will make me more alert mentally.” The corresponding outcome expectancy item states, “How important to you is being more alert mentally at work?” To determine overall expectation, each outcome expectation item was multiplied by the corresponding outcome expectancy item. A composite score was calculated by summing each overall expectation item (outcome expectation score × outcome expectancy score). The final score for the overall expectation item (e.g., outcome expectations–outcome expectancies) ranged from 9 to 225. The outcome expectations–outcome expectancies scale demonstrated high internal consistency reliability in the present study (Cronbach’s α = 0.95).

#### 2.2.5. Sedentary Behavior

The Occupational Sitting and Physical Activity Questionnaire (OSPAQ) was used to assess workplace behavior, particularly occupational sedentary behavior [29]. The OSPAQ is a 3-item self-report measure with questions to assess self-reported weekly working hours, number of days at work per week, and the percentage of a typical workday spent on four behaviors: sitting, standing, light activity, and heavy labor. Total time spent on each behavior is computed by multiplying the reported percentage of time spent on the behavior by the self-reported daily working hours (weekly working hours/number of days at work per week). The OSPAQ has demonstrated sufficient validity to assess self-reported workplace behavior among samples of working adults in Australia, where moderate correlations have been found between self-reported sitting time in the OSPAQ and accelerometer-derived sitting time [29,30].

#### 2.2.6. Leisure Time Physical Activity

Self-reported leisure time physical activity (i.e., accumulated outside of work) was assessed as a potential covariate using the 6-item short form of the previously validated International Physical Activity Questionnaire (IPAQ), which has been shown to be a valid self-report measure of physical activity in adult populations across multiple countries [31]. The IPAQ assesses the number of days and average duration of participation in moderate and vigorous physical activity over the last week. Using the IPAQ scoring guidelines, weekly MET values for walking, moderate, and vigorous physical activity were computed (number of days activity performed × average duration of activity × MET constant for behavior type). Total MET-minutes/week was calculated by summating the MET-minutes/week for walking, moderate, and vigorous activity. Participants were categorized into low (i.e., did not meet moderate or high criteria), moderate (i.e., ≥3 days of vigorous physical activity for ≥20 min OR ≥ 5 days of moderate physical activity or walking for ≥30 min OR ≥ 600 total MET-minutes/week), and high (i.e., ≥3 days of vigorous physical activity AND accumulating ≥1500 MET-minutes OR 7 days of any activity AND accumulating 3000 total MET-minutes/week) physical activity categories.

### 2.3. Data Analysis

Descriptive statistics, including mean and standard deviation for continuous variables, were calculated to describe the sample and study variables. Prior to multi-variable analysis, bivariable analyses were performed to determine differences in occupational sedentary behavior based on demographic and theoretical variables. The Pearson correlation coefficient was used to determine bivariable relationships between continuous demographic and theoretical variables and occupational sedentary behavior, while an independent samples *t*-test and one-way ANOVA were used to determine mean differences in occupational sedentary behavior between groups for the categorical demographic variables. Demographic and theoretical variables demonstrating a significant bivariable relationship with the outcome (i.e., occupational sedentary behavior) were used in the subsequent multi-variable analysis. Hierarchical multiple regression was used to determine the relationship between the demographic and theoretical variables and occupational sedentary behavior, with demographic variables entered in the first block and theoretical constructs entered in the second block to determine the influence of the theoretical constructs above and beyond the demographic variables. The final regression model was assessed to ensure all assumptions were met, including linearity, homoscedasticity, multi-collinearity, and normality. Data were analyzed using SPSS Version 27 (IBM Corp., Armonk, NY, USA) and an alpha value of *p* < 0.05 was considered significant.

## 3. Results

The final sample (*n* = 381) was predominantly white (*n* = 350; 91.9%), female (*n* = 307; 80.6%), married (*n* = 244; 64%), and had a bachelor’s degree or higher (*n* = 305; 80%). There was approximately equal distribution between hourly staff (*n* = 195; 51.2%) and salaried staff (*n* = 186; 48.8%). Mean age was 42.65 (SD = 11.29) years. Average daily occupational sitting time was 5.95 h/8 h workday (SD = 1.30). Most (*n* = 243; 63.8%) participants sat 6 or more hours per 8 h workday. About half (*n* = 194; 50.9%) of participants were classified as moderately physically active, while 23.4% and 25.7% were categorized with low and high leisure time physical activity, respectively. Additionally, approximately one-fourth of employees (*n* = 89; 23.4%) reported currently using a standing desk at work. Detailed demographic information about the sample is presented in Table 1. 

Descriptive information about the SCT variables is presented in Table 2. An independent samples *t*-test analysis showed that employees with a standing desk reported significantly lower hours of sedentary behavior (M = 5.46 vs. M = 6.27) during the workday (*t*[379] = −4.12, *p* < 0.001). A one-way ANOVA analysis showed significant differences in occupational sedentary behavior observed between the low, moderate, and high leisure time physical activity groups, *F*(2, 378) = 11.46, *p* < 0.001. Post hoc analysis using Tukey’s test for multiple comparisons showed that hours of occupational sedentary behavior were significantly lower for the high physical activity group when compared to the moderate (M = 5.51 vs. M = 6.14) and low (M = 5.51 vs. M = 6.58) activity groups. No significant differences in occupational sedentary behavior were observed between groups based on gender, race, education, and job category, *p* > 0.05. Bivariate Pearson correlation analyses showed significant, negative relationships between occupational sedentary behavior and self-efficacy (*r* = −0.297, *p* < 0.001), situational perception (*r* = −0.176, *p* < 0.001), and environment (*r* = −0.148, *p* < 0.001). No significant relationship was found with outcome expectations–outcome expectancies and occupational sedentary behavior (*r* = −0.048, *p* > 0.05).

All demographic and SCT constructs that demonstrated significant univariable relationships with occupational sedentary behavior were included in the final hierarchical linear regression model, with sociodemographic variables (i.e., standing desk ownership, leisure time physical activity level) entered in block 1 and SCT variables (i.e., self-efficacy, situational perception, workplace environment) entered in block 2 (Table 3). The full regression model was significant, R^2^ = 0.133, F(5, 360) = 12.194, *p* < 0.001. In model 1, standing desk ownership (reference: no; β = −0.185, t = −3.68, *p* < 0.001) and leisure time physical activity (MET-minutes/week; β = −0.223, t = −4.42, *p* < 0.001) explained 9.6% of the variance in sedentary behavior (R^2^ = 0.096). In model 2, the SCT constructs led to a statistically significant increase in R^2^ of 0.049, where standing desk ownership (β = −0.139, *t* = −2.75, *p* = 0.006), leisure time physical activity (β = −0.163, *t* = −3.12, *p* = 0.002), and self-efficacy (β = −0.179, *t* = −3.24, *p* = 0.001) explained 13.3% of the variance in sedentary behavior (R^2^ = 0.133).

## 4. Discussion

The present study explored the association between sociodemographic factors and SCT constructs and occupational sedentary behavior among a sample of employees at a large, public university in the south-central United States. Findings from this study showed that average daily occupational sitting time among participants was 5.95 (SD = 1.30) h/8 h workday, which mirrors findings from recent research that has shown employees in desk-based occupations spend a considerable portion of the workday sitting [17,25,32]. Participants in this study who had a standing desk and engaged in high levels of leisure time physical activity reported less sitting time during the workday than those who did not have a standing desk and engaged in low and moderate levels of leisure time physical activity. Significant, negative relationships were also found between occupational sedentary behavior and the SCT constructs self-efficacy, situational perception, and the workplace environment. In the multi-variable regression model, we found that standing desk ownership, leisure time physical activity level, and self-efficacy were significantly associated with occupational sedentary behavior and explained 13.3% of the variance in occupational sedentary behavior among participants.

The aim of this study was to assess the utility of the SCT constructs in explaining occupational sedentary behavior among a sample of university employees. A bivariable relationship was found between situational perception (i.e., social norms), the physical workplace environment, and self-efficacy, where all three SCT constructs demonstrated a significant, negative relationship with occupational sedentary behavior. Additionally, in the multi-variable model, self-efficacy retained a significant, negative relationship with occupational sedentary behavior, whereas the other SCT variables did not. SCT as a theoretical framework emphasizes the importance of considering the reciprocal relationship between the physical/social environment, cognitive factors, and health behavior to encourage and support health behavior change [22]. To date, most theory-based workplace interventions to reduce sedentary behavior have utilized SCT [10,18], which is most likely due to the increasing emphasis on utilizing multi-component approaches that include educational/behavioral approaches and environmental support to change workplace sedentary behavior [10,11]. The findings from this study support the incorporation of educational/behavioral strategies in multi-component interventions to address SCT constructs, most notably self-efficacy.

SCT posits that self-efficacy is a major determinant of health behavior change, and the findings from this study support this claim regarding reducing sedentary behavior in the workplace [22]. In the bivariable and multi-variable analysis, a significant, negative association was found between self-efficacy to reduce sedentary behavior during the workday and employees’ sitting time during the workday. Findings from this study suggest that increases in an employee’s self-efficacy, or confidence, to reduce sitting time during the workday are associated with reduced sitting time during the workday. Self-efficacy beliefs originate from four sources: mastery experiences (i.e., demonstrating success in mastering a behavior), vicarious experiences (i.e., observations of peers and colleagues performing the behavior), verbal persuasion (i.e., receiving verbal encouragement to master the behavior), and emotional/physiological states (creating emotional and physiological states that support a behavior) [33]. Health education and promotion practitioners and researchers designing workplace interventions should consider incorporating strategies to increase self-efficacy to reduce sitting time during the workday in addition to other intervention strategies, such as the provision of standing desks or other educational approaches.

Potential intervention strategies may be informed by previous research that has explored physical activity and sedentary behavior change among adults. A meta-analysis of physical activity interventions found that the behavior change techniques associated with higher physical activity effect sizes included action planning, reinforcement of behavior change, assessments of behavioral progress, provision of instructions, facilitation of social comparison, and time management [34]. Additionally, potential intervention strategies may be informed by previous quantitative and qualitative research assessing sedentary behavior change in the workplace, where participants have expressed a need for activity monitoring, reminders, educational/instructional programming, social comparison, and accountability [11,35,36,37,38], all of which may help facilitate increases in self-efficacy to reduce occupational sedentary behavior. Behavior change techniques identified in previous physical activity and sedentary behavior research may be useful to increase self-efficacy to reduce sedentary behavior among working adults and should be explored in future interventions.

In addition to findings relevant to SCT constructs, leisure time physical activity level was found to be significantly associated with occupational sedentary behavior among participants in this study. Specifically, individuals with high leisure time physical activity levels demonstrated significantly lower occupational sedentary behavior than those with moderate and low/no leisure time physical activity. In the multi-variable model, leisure time physical activity (MET-minutes/week) demonstrated a significant, negative association with occupational sedentary behavior. This finding is important, as recent epidemiological research has suggested a dose–response relationship between sedentary behavior and mortality, where the highest mortality risk has been observed among adults who sit more and move less [5,6]. In addition to addressing SCT antecedents of behavior change in future interventions, workplace health promotion practitioners and researchers should consider targeting adults with high sedentary behavior and low leisure time physical activity in future workplace interventions.

### Limitations

There are limitations to this study that should be considered when interpreting the study findings. First, all data were collected using self-report measures, which increases the likelihood of potential response biases, including social desirability. This may have impacted the findings in the analysis. Additionally, the OSPAQ, which was used to assess occupational sedentary behavior, could have underestimated the amount of time spent sitting during the workday. Research has shown that sitting is perceived as an “invisible behavior”, where participants may have more difficulty recalling sitting time on a self-report instrument [39]. However, all self-report measures used in this study demonstrated evidence of reliability and validity in this study and in previous research, which supports the utility and appropriateness of these measures to assess the study variables. Second, study participants were recruited from one workplace, a large, public university in the south-central United States, which limits the generalizability of the study findings to other employees and workplaces. Additionally, the sample was predominantly female and white, further limiting the generalizability of the findings to other employee sub-groups. However, the sample was representative of the larger employee population at the site where data were collected. Third, data gathered in this study were collected using a cross-sectional study design, which limits any interpretations regarding causality and directionality. Future studies should implement the use of objective measures of behavior, include more diverse samples with respect to sociodemographic characteristics and workplace setting, and incorporate longitudinal study designs to overcome the limitations in the present study.

## 5. Conclusions

The current study supports the utility of SCT constructs, notably self-efficacy, to explain occupational sedentary behavior among a sample of working adults in a university setting. The present study found a significant, negative relationship between SCT constructs, including self-efficacy, situational perception, and the environment, and occupational sedentary behavior. To date, most interventions that address occupational sedentary behavior are not informed by a health behavior change theoretical framework [17,32]. Findings from this study support the potential use of intervention strategies to change theoretical antecedents of behavior change such as through the inclusion of educational/behavioral strategies in future multi-component workplace interventions to address sedentary behavior. Behavior change techniques to modify self-efficacy, such as activity monitoring, sending reminders, educational programming, and social comparison, are supported by previous quantitative and qualitative research on occupational sedentary behavior and may be beneficial to incorporate into future workplace interventions. Future research should explore alternative theoretical frameworks to determine additional factors associated with sedentary behavior. Newer physical activity theoretical frameworks, such as the theory of effort minimization in physical activity (TEMPA), suggest that the disparity between the intention to be physically active (i.e., reduce sedentary behavior) and actual engagement in physical activity may be due to the automatic tendency to seek effort minimization, which is achieved through increased time in sedentary behavior [40]. Frameworks such as the TEMPA may be helpful to describe additional antecedents of sedentary behavior not addressed in the present study. Future workplace interventions should continue to use multi-component approaches combining educational/behavioral strategies with environmental modifications to support sedentary behavior change among working adults.

## Figures and Tables

**Table 1 ijerph-20-06476-t001:** Demographic characteristics of the sample (*n* = 381).

Demographic Characteristic	*n* (%) ^1^	*M* (*SD*)
Employment Classification		
Hourly Staff	194 (51.2%)	
Salaried Staff	186 (48.8%)	
Gender Identity		
Female	307 (80.6%)	
Male	73 (19.2%)	
Racial Identity		
White	350 (91.9%)	
Other	40 (10.5%)	
Education Level		
Less than College/Some College	75 (19.7%)	
Bachelor’s Degree	150 (39.4%)	
Graduate/Professional Degree	155 (40%)	
Leisure Time Physical Activity Category		
Low	89 (23.4%)	
Moderate	194 (50.9%)	
High	98 (25.7%)	
Standing Desk Ownership		
Yes	89 (23.4%)	
No	292 (76.6%)	
Sedentary Behavior (h/8 h workday) ^2^		5.95 (1.30)
<6 h/8 h workday	138 (36.2%)	
≥6 h/8 h workday	243 (63.8%)	

^1^ Frequencies represent the valid percent. Participants were not required to answer all survey items. Participants could select more than one racial identity. ^2^ h/8 h workday was standardized to an 8 h workday to account for variability in hours worked/day among participants.

**Table 2 ijerph-20-06476-t002:** Descriptive statistics for the social cognitive theory construct variables.

Variable	Possible Score Range	Observed Score Range	Mean ± SD	Cronbach’s Alpha
Self-Efficacy	8–40	8–40	22.10 ± 8.53	0.90
Outcome Expectations–Expectancies	9–225	33–225	158.95 ± 42.98	0.95
Situational Perception	8–40	15–40	31.13 ± 4.86	0.80
Environment	3–16	3–12	10.01 ± 2.02	0.85

SCT variables were operationalized based on the SCT framework from Bandura (2004) [21].

**Table 3 ijerph-20-06476-t003:** Hierarchical multiple linear regression explaining occupational sedentary behavior.

Variables	Model 1		Model 2	
	B	β	B	β
Constant	6.61 **		8.35 **	
Standing Desk Ownership	−0.704 **	−0.185	−0.526 *	−0.139
MET-min/week of Leisure Time Physical Activity	<0.001 **	−0.223	<0.001 *	−0.163
Self-Efficacy			−0.034 **	−0.179
Situational Perception			−0.023	−0.069
Workplace Environment			−0.033	−0.042
R^2^	0.091		0.133	
ΔR^2^	0.096		0.049	

B (unstandardized coefficient)**;** β (standardized coefficient); * *p*-value <0.05; ** *p*-value < 0.001; adjusted R^2^ of the final model = 0.133.

## Data Availability

Data are available upon request due to privacy/ethical restrictions.

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
