# Peer review of "A Social Cognitive Assessment of Workplace Sedentary Behavior among a Sample of University Employees"

_ijerph, 2023, doi:10.3390/ijerph20156476_

Round 1

Reviewer 1 Report

Dear authors,

I sincerely appreciate the rigor and dedication you have put into the development of your manuscript. Your study, based on social cognitive theory, has the potential to make significant contributions to our understanding of sedentary behavior. However, in order to further enhance the quality of your work, I would like to offer the following suggestions:

Abstract: is quite informative, however, the authors must state the type of study design.

2.1. Sample and Recruitment: I suggest detailing the number of participants before and after applying the inclusion or exclusion criteria, and detailing the number of participants eliminated according to the criteria.

2.2 Measure: It is important to provide details about the context, such as the population and location in which the reported internal consistency has been found in your study. I would appreciate it if you could consider this comment in relation to all the mentioned measurement instruments.

2.2.1 Self-Efficacy: The original scale was designed to assess self-efficacy in the context of exercise, which significantly differs from sedentary behavior. Therefore, the adaptation process undertaken requires a more detailed report. In this study, some items from the original scale were selected and subsequently modified to specifically address sedentary behavior. Hence, it is important to reported the psychometric analyses that support the validity of the adapted scale used. Additionally, it is relevant to indicate if a cognitive test of the modified instrument was conducted and to describe the modifications that arose from its application. If this procedure of adaptation and validation of the self-efficacy scale, presented in this study, has been previously published elsewhere, the corresponding reference should be provided.

However, I suggest reconsidering the measure of self-efficacy in sedentary behavior, since if we take into account that self-efficacy, according to Albert Bandura's social cognitive theory, refers to a person's belief in their ability to successfully carry out a specific task or achieve a desired goal, consequently, what was done was to measure the skills and competencies that the participants believed they had to be sedentary. My question is: is this the information that the authors really wanted to get in their study?

line 155-156: If the reported internal consistency refers to the original exercise self-efficacy scale, it is important to note that in this study we are using an adapted scale that specifically focuses on self-efficacy for sedentary behavior. Therefore, the reported information on the internal consistency of the original scale is not relevant.

2.3 Data Analysis: 

line 228: I suggest that this statement be left at the end of this paragraph (Data were analyzed using SPSS Version 27)

line 229-230: As I mentioned in previous comments, this information (with all values ≥ .70 deemed acceptable) must be detailed by instrument in section 2.2. Measures

line 230-231: There is the possibility of performing a data imputation process in case of missing values. I suggest considering this option to avoid sample loss.

line 232-233: This information should be detailed and reported in section 2.1. Sample and Recruitment. If adequately described in that section of your manuscript, there is no need to report this information here, where you only should provide details about the statistical procedures conducted

3. Results

line 255-256: In Table 1 does not include data on the amount of sedentary behavior hours of the participants.

Table 1: Since sedentary behavior is one of the main variables in this study, it would be interesting to have a more detailed description of this behavior in the studied sample. For example, descriptive information based on commonly used classifications to determine if a person is sedentary or not (>8 hours per day according to Patterson et al.1) could be provided in Table 1, presenting the information as a total, and also according to the group of sedentary and non-sedentary participants. Additionally, the heterogeneity of the sample in relation to gender is also noteworthy, as it may influence the results. Therefore, this variable should be taken into account in the analyses, possibly as a covariate within the reported model.

1Patterson R, McNamara E, Tainio M, de Sá TH, Smith AD, Sharp SJ, et al. Sedentary behaviour and risk of all-cause, cardiovascular and cancer mortality, and incident type 2 diabetes: a systematic review and dose response meta-analysis. Eur J Epidemiol [Internet] 2018; 33 (9): 811-29. Available from: http://www.ncbi.nlm.nih.gov/pubmed/29589226.

Table 2:

line 267-270: I suggest that the information provided here be reported in the Materials and Methods section. Specifically, the information regarding internal consistency using Cronbach's alpha coefficient should be reported for each instrument, as mentioned in previous comments. As for the measures of kurtosis and skewness of the data distribution for each variable, it would be sufficient to mention in the statistical analysis section how the assumption of normality was tested and whether it was met. Also, I suggest including the descriptive statistics of the mean and standard deviation for each variable related to social cognitive theory in Table 1 and removing Table 2.

Table 3: It is incorrect to report a value of 0.000 from a statistical perspective. Instead, it should be reported as <0.001. In the caption, italic formatting has been used to report the p-value. I believe this is the most appropriate way to do it, so I suggest standardizing this aspect throughout the manuscript.

Author Response

We appreciate the reviewer's feedback and feel that the feedback has strengthened the manuscript. Please find our response to the reviewer's comments in the attached document.

Reviewer 2 Report

General comment:

Thank you for the opportunity to review the manuscript titled "A Social Cognitive Assessment of Workplace Sedentary Behav-2 ior among a Sample of University Employees”. I believe that this article will be of great interest to readers as well as researchers in the field. This study examined the utility of SCT constructs to explain occupational sedentary behavior among a sample of university employees. The authors used a two-stage modeling approach while focusing on different temporal levels (hours, daily, burst). The results suggest that self-efficacy may be an important factor in explaining variation in 23 occupational SB. Below are some major and minor comments, which could be acknowledged to improve clarity.

Major comments:

1.       Did the authors formulate specific hypotheses before the statistical analyses? If yes, please add this to the instruction. If not, please indicate that the analyses are based on exploratory analyses.

2.       I was wondering whether the used questionnaire may underestimate sedentary time. Previous studies have shown that sitting can be seen as an invisible behavior (i.e., with a less accessible cognitive representation of seated activities; https://doi.org/10.1186/s12966-019-0851-0). I would recommend discussing this issue, compared to a questionnaire with more items (https://doi.org/10.1123/jpah.7.6.697).

3.       P. 9: lines 357-368: The finding that PA is related to occupational SB can be discussed under consideration of current analytical approaches such as compositional data analyses. Both SB and PA are strongly correlated from a time perspective (that means while you are being physically active, at the same time you can´t be sedentary).

4.       It would be interesting to discuss the general tendency of spending longer time in sedentary behavior. A recently published article (10.1249/JES.0000000000000252) brings up the Theory of Effort Minimization in Physical Activity. Relevant constructs from this theory may also be valuable to explain occupational SB.

Minor comments:

1.       P.1 lines 38-40: I recommend adding the 2020 WHO-Guidelines.

2.       P.4 line 179: The citation Blunt & Hallam should be indexed as [??]

3.       P.6 line 252: Would the author expect different results for a more balanced sex distribution?

4.       P.6 line 256: Could the authors provide some information about the range of sitting time? 

Author Response

(The authors gave the same response as above.)
